# Comparative Analysis of Perivascular Adipose Tissue Attenuation on Chest Computed Tomography Angiography in Patients with Marfan Syndrome: A Case–Control Study

**DOI:** 10.3390/diagnostics15060673

**Published:** 2025-03-10

**Authors:** Domenico Tuttolomondo, Francesco Secchi, Nicola Gaibazzi, Nathasha Samali Udugampolage, Alessandro Pini, Massimo De Filippo, Pietro Spagnolo, Rosario Caruso, Jacopo Taurino

**Affiliations:** 1Cardiology Unit, Azienda Ospedaliero-Universitaria di Parma, Via Antonio Gramsci 14, 43126 Parma, Italy; d.tuttolomondo@hotmail.it (D.T.); ngaibazzi@gmail.com (N.G.); 2Department of Biomedical Sciences for Health, Università degli Studi di Milano, Via Mangiagalli 31, 20133 Milano, Italy; francesco.secchi@unimi.it (F.S.); rosario.caruso@unimi.it (R.C.); 3Unit of Cardiovascular Imaging, IRCCS MultiMedica, Via Milanese 300, 20099 Milano, Italy; 4Cardiovascular-Genetic Center, IRCCS Policlinico San Donato, San Donato Milanese, 20097 Milano, Italy; alessandro.pini@grupposandonato.it (A.P.); jacopo.taurino@grupposandonato.it (J.T.); 5Department of Medicine and Surgery, Section of Radiology, University of Parma, Maggiore Hospital, Via Gramsci 14, 43125 Parma, Italy; massimo.defilippo@unipr.it; 6Unit of Radiology, IRCCS Policlinico San Donato, San Donato Milanese, 20097 Milano, Italy; pietro.spagnolo@grupposandonato.it; 7Health Professions Research and Development Unit, IRCCS Policlinico San Donato, San Donato Milanese, 20097 Milano, Italy

**Keywords:** Marfan syndrome, perivascular adipose tissue attenuation, computed tomography angiography, vascular inflammation

## Abstract

**Background**: Marfan syndrome (MFS) is a rare autosomal dominant disorder affecting connective tissues due to mutations in the fibrillin-1 gene. These genetic changes often result in severe cardiovascular conditions, including asymptomatic thoracic aortic dilation potentially leading to dissection or rupture. Perivascular adipose tissue attenuation (PVAT) observed on computed tomography may serve as a marker of localized inflammation and indicate early histopathological changes in the vascular walls of MFS patients compared to healthy individuals. **Objective**: This study aimed to compare PVAT values between patients with MFS and healthy controls in order to explore whether MFS patients show higher PVAT secondary to these histopathological abnormalities. **Methods**: This case–control study assessed PVAT on ascending aorta through computed tomography angiography (CTA) in 54 genetically confirmed MFS patients and 43 controls with low ischemic risk, excluding those with known aortic aneurysms. **Results**: PVAT analysis revealed significant differences between the MFS patients and healthy controls (−70.6 HU [−72.6 HU to −68.5 HU] versus −75.1 HU [−77.1 HU to −73.1 HU], *p* = 0.002), suggesting potential early vascular changes in the MFS group. **Conclusions**: The findings underscore the potential diagnostic role of PVAT in patients with genetically confirmed MFS but normal ascending aorta diameter.

## 1. Introduction

Marfan syndrome (MFS, OMIM # 154700) is a rare disease of autosomal dominant inheritance with an estimated prevalence of 1:5000 people in the general population, with no differences in gender or ethnicity [1]. MFS is a complex systemic disorder characterized by significant inter- and intrafamilial variability primarily affecting connective tissues; this variability is due to heterozygous mutations in the fibrillin-1 (FBN1) gene, located on the long arm of chromosome 15 (15q21.1), which encodes the extracellular matrix protein [2]. Impairment in FBN1 protein synthesis, secretion, or incorporation into the extracellular matrix typically results in major abnormalities in the cardiovascular, ocular, and musculoskeletal systems [3]. The FBN1 gene, a major component of the microfibril core, plays a crucial role in developing a normal extracellular matrix scaffold. Pathogenic variants in FBN1 typically result in suboptimal elastic fiber composition, usually compensated by an excessive deposition of collagen and proteoglycans, which contributes to increased stiffness and progressive weakening of the extracellular matrix [4,5,6].

Cardinal phenotype, which is typical but highly variable, includes the presence of aortic dilation evolving in thoracic aortic aneurysm (TAA), eye lens dislocations (ectopia lentis) and alterations of the physiological curves of the spine, as well as chest deformities (i.e., pectus excavatum). The prognosis of MFS is related to the severity of the phenotype: cardiovascular complications, such as progressive asymptomatic aortic dilation, are the most threatening, potentially leading to dissection or rupture as well as mitral and tricuspid valve prolapse and enlargement of the proximal pulmonary artery [7,8].

Diagnosis refers to the international revised Ghent criteria of 2010 and considers the presence of (A) aortic root dilation/dissection with ectopia lentis; (B) aortic root dilation/dissection with FBN1 mutation; and (C) ectopia lentis with FBN1 mutation [9]. The arousal of clinical symptoms is significantly changeable between affected individuals, ranging from severe cardiovascular complications present at birth (neonatal form) to patients developing manifestations later in life. The average lifespan for patients left untreated is typically around 40 years [10]. However, advancements in research and early diagnosis and management have increased patients’ life expectancy in the last 30 years [3]. Syndromic manifestations in MFS patients require multidisciplinary evaluation, management and treatments. For cardiac evaluation, trans-thoracic echocardiogram is the gold standard for clinical assessment of the aortic diameters such as sinuses of Valsalva, the sino-tubular junction and the ascending aorta at the level of the right pulmonary artery [11]. Together with chest computed tomography angiography (CTA), the gold standard imaging test for thoracic aortic disease in adult populations, these approaches yield data regarding the aneurysm’s location and dimensions, as well as the degree and classification of the dissection, adjacent hemorrhagic regions, and other pathological complications resulting from the dissection. They comprehensively evaluate thoracic aorta aneurysms in terms of morphology, extent, thrombosis, relationship to adjacent structures and signs of rupture [12]. Therefore, regular imaging of TAAs should be carried out in patients with asymptomatic aortic root and ascending aortic aneurysms or those with aortic dissection to assess the need and time for preventative surgery. Moreover, CTA is also recommended for first-degree relatives as it is essential to detect unrecognized asymptomatic aortic aneurysms [13]. Currently, preemptive surgical intervention for TAAs is the most successful technique for preventing aortic rupture and dissection, demonstrating successful TAA repair and superior long-term survival rates compared to age- and gender-matched controls.

Perivascular adipose tissue (PVAT) attenuation on CTA imaging is recognized as a marker of localized vascular inflammation and reflects histopathological changes in perivascular adipose tissue [14,15]. PVAT is a highly dynamic and metabolically active tissue that envelopes blood arteries [15]. Ascending aortic perivascular adipose tissue is a distinct segment of perivascular adipose tissue that envelopes the aortic root, the ascending aorta and the origin of the coronary arteries, making it the closest perivascular adipose tissue to the heart. Previously applied in coronary arteries and other vascular beds, PVAT analysis via CTA provides valuable insights into focal points and non-morphological changes [16,17,18,19,20,21]. It has been demonstrated that coronary vessels exert paracrine effects on the surrounding adipose tissue; this leads to morphological changes in adipocytes, both in terms of their size and lipid content [22]. Such morphological changes can be detected through the measurement of PVAT on coronary arteries; it may also be useful to improve cardiac risk prediction through coronary CTA since it provides a quantitative measure of coronary inflammation. Peri-coronary adipose tissue attenuation on CTA is able to predict allograft rejection and cardiovascular events in heart transplant recipients and stent failure in patients undergoing percutaneous coronary intervention [23,24]. Inflammation plays a key role in the pathogenesis and progression of ischemic cardiovascular disease. The use of immunomodulatory drugs, such as Canakinumab and Colchicine, in patients with chronic coronary syndrome reduces the risk of recurrent cardiovascular events [25,26,27].

PVAT levels around ascending aorta aneurysms are typically higher in MFS patients compared to those with a normal aortic diameter, independently of the volume of perivascular adipose tissue [28]. In patients without MFS but with ascending aorta aneurysms, PVAT has been linked to a progressive replacement of perivascular adipose cells with fibrotic tissues and with degeneration of the elastic fiber of the vascular wall [29]. However, there is paucity of evidence in the literature analyzing PVAT levels in MFS and their potential correlations with TAA development and aortic inflammation.

This study hypothesizes that vascular tissue alterations in MFS patients might lead to measurable changes in perivascular adipose tissue, detectable through PVAT analysis on CTA, before the development of ascending thoracic aorta aneurysms. Furthermore, a prior case series reports very high PVAT values in the ascending thoracic aorta, −61.7 Hounsfield units (HU), in three phenotype-negative asymptomatic patients with a genetic diagnosis of MFS [30].

The current study investigated PVAT on CTA in MFS patients compared to healthy controls, with both groups showing normal and comparable ascending aorta mean diameter. Therefore, this study aimed to compare PVAT values between these two groups, to explore whether MFS patients show higher PVAT secondary to the abovementioned histopathological abnormalities.

## 2. Materials and Methods

### 2.1. Patients and Study Design

This study was conducted in a Cardiovascular-Genetic Center for rare diseases at a large teaching hospital in Northern Italy (Istituto di Ricovero e Cura a Carattere Scientifico—IRCCS—Policlinico San Donato, Milan, Italy). We retrospectively enrolled two groups of participants with normal ascending aorta diameters after verifying adherence to predefined inclusion and exclusion criteria. The first group consisted of 54 phenotype-negative asymptomatic patients with genetically confirmed MFS. MFS patients underwent CTA between October 2007 and December 2022. The second group (control group) included 43 consecutive, healthy subjects at low risk of ischemic cardiovascular disease. Controls underwent CTA due to atypical chest pain between February 2022 and October 2022. While MFS diagnoses in the patient group were confirmed genetically, the controls were selected from the general population undergoing CTA for atypical chest pain, normal ascending aortic diameter and a low pre-test probability of coronary artery disease. Although there was no clinical suspicion of MFS in the control group, we further ensured the likely absence of MFS in the control group by excluding any individuals with a family history of MFS, aortic aneurysm, rupture, or dissection.

#### 2.1.1. Inclusion Criteria

For the MFS group, inclusion required that a CTA had to be performed after receiving a genetic diagnosis of MFS, in asymptomatic patients, as per the protocol in our center. Conversely, participants in the control group were selected based on having undergone aortic CTA due to atypical chest pain and having a low pre-test probability of coronary artery disease. All participants, regardless of group, were required to have no detectable aortic aneurysms or dissections and no significant coronary artery disease (defined as any luminal coronary artery stenosis more than 30%), as well as no known cardiovascular risk factors, including hypertension, diabetes mellitus, obesity, dyslipidemia and tobacco use. Moreover, candidates with a history of ischemic cardiovascular disease, previous acute myocardial infarction, coronary revascularization, chronic coronary syndrome or prior heart or vascular surgery, chronic inflammatory or infectious diseases and any current or previous malignancy were excluded from the study. These criteria were designed to ensure a homogeneous study population and to minimize confounding variables that could impact the assessment of PVAT and its potential association with MFS.

#### 2.1.2. Exclusion Criteria

Participants under the age of 18 were excluded in order to focus the study on adult populations. For the control group, individuals with a family history of MFS, aortic aneurysm, rupture, or dissection were also excluded to avoid confounding genetic influences. Any potential participants who showed signs of lung disease on CTA scans were not included in order to avoid confounding factors on PVAT measurement. Lastly, anyone who refused to participate in the study was excluded, maintaining ethical standards for voluntary participation in clinical research.

### 2.2. High-Resolution Computed Tomography (HRCT)

All studies were conducted using a second-generation 128-slice dual-source computed tomography (CT) system (Definition Flash, Siemens Healthcare, Nurnberg Germany). Scans were performed in the cranio-caudal direction with a prospective electrocardiogram-triggered protocol. Contrast-enhanced CT scans were acquired from the thoracic inlet to the pubic symphysis. The scanning parameters for both groups were as follows: slice collimation of 128 × 0.6 mm with a z-flying focal spot, a gantry rotation time of 280 ms, a pitch of 3.2, and a tube voltage of 100 kV (low-iodine group) or 120 kV (high-iodine group). A bolus tracking technique was employed to monitor the contrast material injection, with the region of interest located in the middle of the descending aorta and a threshold of 100 HU, followed by a 4-s delay. The contrast medium was administered at a dose of 1 mL per kg of body weight, followed by a 30 mL saline flush. The injection rate for both the contrast medium and saline was 4 mL/s for all subjects. CTA images were reconstructed using a conventional filtered back projection algorithm with a medium smooth kernel (B26f), which is optimized for cardiac imaging. For the low-iodine group, images were also reconstructed using a sinogram-affirmed iterative reconstruction algorithm (SAFIRE, Siemens Healthcare, Erlangen, Germany) with the corresponding vascular kernel (I26f). The iterative reconstruction algorithm was applied with a medium strength setting of 1, as recommended by the manufacturer.

### 2.3. Peri-Coronary Adipose Tissue Attenuation

To measure the PVAT in the ascending aorta, we used a software package (Aquarius Workstation^®^ V.4.4.13, TeraRecon Inc., Foster City, CA, USA) that enabled us to trace volume samples. Figure 1 illustrates the step-by-step post-processing method applied to measure the mean PVAT systematically to all CT scans. Specifically, after reorienting the imaging planes to obtain the long and short axes of the ascending aorta, the short-axis plane was used to draw a volume sample, interpolated between two traced circles. The first circle was drawn at the level of the aortic valve/cusps, and the second was placed 40 mm distally along the aortic course. Both circles were traced with a radius equal to the aortic radius at each specific level, plus an additional 10 mm (measured as the radial distance from the outer aortic wall). Perivascular adipose tissue within this volume sample was automatically quantified by the software based on the attenuation histogram of adipose tissue, within a range of −190 HU to −30 HU, as previously described [9]. To account for technical variations over time, we adjusted the PVAT measurements for the voltage used in the CT scan. If 100 kV was used instead of the standard 120 kV, the mean HU value was corrected by dividing by a factor of 1.11485. The software automatically measured the adipose tissue contained within the drawn volume sample, which was shaped like an irregular hollow cylinder. This measurement provided both the total fat volume and the mean PVAT ± standard deviation (SD).

Figure 1 shows proper alignment of the aortic valve plane on CTA.

### 2.4. Technical Steps Required to Measure Perivascular Adipose Tissue Attenuation of the Proximal Ascending Thoracic Aorta on Chest Computed Tomography Angiograms

Figure 2 shows the selection of the region of interest of the ascending thoracic aorta on CTA.

Figure 3 shows the perivascular adipose tissue of the proximal ascending thoracic aorta.

Figure 4 shows the 3D reconstruction of perivascular adipose tissue of the proximal ascending thoracic aorta.

### 2.5. Statistical Analyses

Variables were expressed as numbers and percentages, mean and standard deviation or median and 25–75% percentile range (lower–upper quartile), as appropriate. The characteristics of participants were then compared between the MFS and control groups with the chi-square test, a *t*-test or the Mann–Whitney test, as appropriate, depending on the variable and its distribution.

Analyses were performed with the MedCalc statistical package (version 19.4, Ostend, Belgium), considering *p* values < 0.05 as statistically significant.

## 3. Results

### 3.1. Sample Characteristics

This study included 97 participants with an average age of 43.4 years (standard deviation, SD = ±11 years); of these, 40 individuals (41%) were male. The cohort was divided into two groups: 54 patients with an MFS diagnosis formed the case group, while the remaining 43 participants served as the healthy control group. Both groups were selected based on the absence of cardiovascular risk factors, including hypertension, diabetes mellitus, obesity, dyslipidemia and tobacco use, as well as previous acute myocardial infarction, coronary revascularization and chronic coronary syndrome, aligning with the study’s inclusion criteria. None of the participants has aortic aneurysm or significant coronary artery disease (coronary artery stenosis more than 30%). Detailed clinical characteristics of all participants and a comparative analysis between the MFS and control groups are shown in Table 1. Furthermore, there were no significant differences regarding age (42 ± 13 versus 46 ± 8 years, *p* = 0.058), gender (21% versus 19%, *p* = 0.599) or the diameter of the ascending aorta (33.4 ± 3.9 mm versus 33 ± 3.1 mm, *p* = 0.695) between the two groups.

### 3.2. Comparisons

PVAT of the ascending aorta on CTA was higher in the MFS group compared with the control group (−70.6 HU [−72.6 HU to −68.5 HU] versus −75.1 HU [−77.1 HU to −73.1 HU], *p* = 0.002). No differences in terms of gender were detected in participants.

## 4. Discussion

This case–control study represents the first proof-of-concept exploration of PVAT on CTA as a potential marker of TAA progression in a population with MFS. Despite genetic testing currently being included among the gold standard techniques in MFS diagnosis and determining the timing for clinical evaluation and follow-up, it is typically expensive and frequently performed at advanced stages of the disease [6]. Conversely, CTA plays a critical role in evaluating aortic abnormalities, particularly enlargements aimed at preventing the severe complications and cardiovascular comorbidities associated with MFS. Our findings show that PVAT values measured just around the ascending aorta are significantly elevated in MFS patients compared to healthy controls, suggesting heightened inflammatory activity and histopathological changes in the vascular wall, associated with this genetic syndrome.

Considering previous studies on PVAT supporting the notion that local adipose tissue inflammation might serve a function in aortic remodeling in patients with aortic aneurysms and coronary artery diseases, conditions with genetically induced alteration of the structural composition of the aortic vessel might increase the exposure to localized vascular inflammation and remodeling: while the study protocol was not originally designed to determine prognostic indicators, the results may suggest that higher PVAT could potentially be applied in the risk stratification of aortic complications in relation to the severity of the condition in MFS patients. It is crucial to note that elevated PVAT may not solely be an indicator of inflammation, but could also represent various histopathological changes in the peri-aortic tissues. This hypothesis is supported by prior research on patients with ascending aorta aneurysms who have undergone surgical interventions, where PVAT measured on CTA was associated with unique tissue characteristics [16,17].

As described in the literature, localized inflammation in the coronary arteries, investigated by PVAT on coronary computed tomography, predicts the risk of cardiovascular death and death from all causes [31,32,33,34]. While inflammation plays a key role in abdominal aortic aneurysms, evidence supporting the role played by inflammation in the pathogenesis of thoracic aneurysms, in particular for syndromic and non-syndromic TAAs, is scarcely described in the literature. In this scenario, the application of this method in vascular districts such as the thoracic aorta should be considered. Unlike in coronary arteries, only a few studies have tested this method in other vessels, although the results seem to be interesting [28,35,36,37].

The 2014 guidelines of the European Society of Cardiology regarding patients with thoracic aortic aneurysms address an elevated risk of cardiovascular events [11], primarily unrelated to the aneurysm itself, but potentially associated with shared risk factors (e.g., smoking or hypertension) and mechanisms (e.g., inflammation), along with a heightened risk of cardiovascular comorbidities at the time of aneurysm diagnosis. Notwithstanding considerable progress in managing life-threatening complications related to the disease, including the surgical correction of thoracic aortic aneurysm and dissection, it is still essential to apprehend and identify non-invasive markers for MFS and genetically related TAAs, for diagnosis and prognosis, and for novel therapeutic targets for MFS. An understanding of the complex pathomolecular mechanisms of the condition is essential to meet these objectives.

Moving forward, an observational prospective study to longitudinally track PVAT measurements in MFS patients will be of key importance, not only in terms of these measurements, but especially in terms of clinical outcome (i.e., dilatation, dissection and need for prophylactic surgery). Such research would aim to verify whether early detection of increased PVAT could represent a reliable indicator of vascular inflammation as well as impending severe cardiovascular complications. Moreover, such longitudinal evaluation would emphasize the predictive value of PVAT and identify patients who need more thorough clinical follow-up. In this sense, it is our intention to continue the analysis and study in longitudinal terms. The ultimate goal is to refine individual risk assessments, facilitating a personalized approach to the follow-up and management of MFS, according to precision medicine.

However, this study is not without limitations. The study’s retrospective nature may introduce selection biases, as the cases and controls were not prospectively followed. Additionally, the absence of genetic testing in the control group means that subclinical cases of MFS with normal aortic diameter could not be definitively excluded, although efforts were made to minimize this possibility. Furthermore, the study only measured PVAT in individuals without significant cardiovascular diseases or risk factors, which might limit the generalizability of the findings to all MFS patients. Despite these limitations, the study has several strengths. Using a well-defined case–control design, it is the first to quantitatively assess PVAT in MFS patients compared to a control group with normal aortic diameters. The use of advanced imaging techniques and the comprehensive characterization of the study population enhance the reliability of the findings. Additionally, the significant differences in PVAT found between the groups provide valuable insights that could lead to the development of non-invasive markers for predicting vascular complications in MFS patients.

## 5. Conclusions

This case–control study has confirmed for the first time the hypothesis that PVAT levels around the ascending aorta, as measured by CTA, are significantly higher in adult patients with MFS compared with healthy individuals, despite both groups presenting with normal thoracic aortic diameters. This increased PVAT may reflect intrinsic histological changes in the elastic fibers of the vascular wall in MFS patients, as described in the literature, and might be associated with impairment in FBN1 protein synthesis and extra cellular matrix remodeling: these changes could potentially occur before any signs of aortic enlargement detectable via TTE. Considering that these findings are encouraging, extensive and comprehensive validation in large patient populations is essential to establish their clinical significance and therapeutic potential to fully establish the diagnostic and prognostic value of PVAT in thoracic aortic aneurysms in MFS patients.

Subsequent studies will be crucial in assessing whether PVAT can be effectively applied in detecting and predicting major vascular complications, potentially serving as an early biomarker for proactive disease management and follow-up in this population.

## Figures and Tables

**Figure 1 diagnostics-15-00673-f001:**
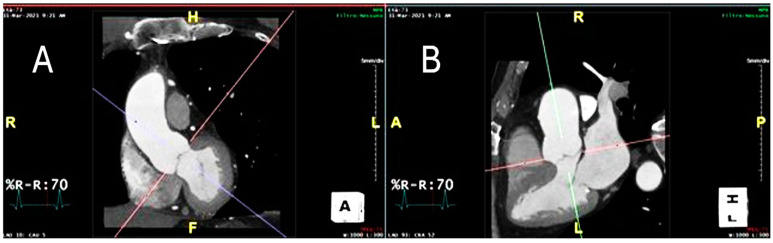
The figure shows proper alignment of the planes in the coronal (**A**) and the sagittal (**B**) aortic valve plane.

**Figure 2 diagnostics-15-00673-f002:**
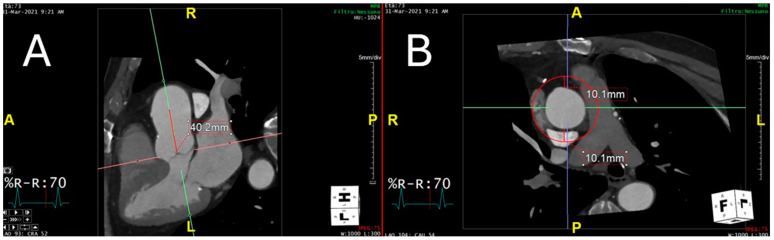
The proximal ascending thoracic aorta was selected for a length of 40 mm from the aortic valve plane (**A**), and a region of interest plus 10 mm radial to the vessel was then selected (**B**).

**Figure 3 diagnostics-15-00673-f003:**
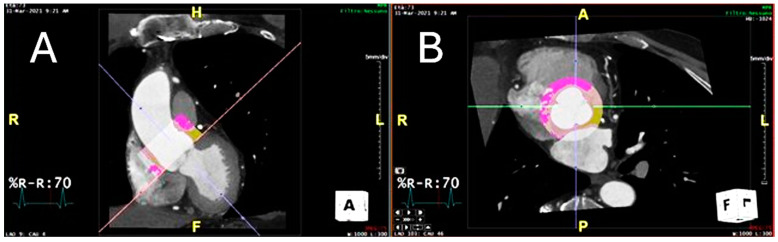
The figure shows the aortic perivascular tissue at the level of (**A**) the ascending aortic coronal plane and at the level of (**B**) the aortic valve plane and the coronal plane. In yellow is the perivascular tissue with a density between −30 and −190 Hounsfield units, which corresponds to the adipose tissue.

**Figure 4 diagnostics-15-00673-f004:**
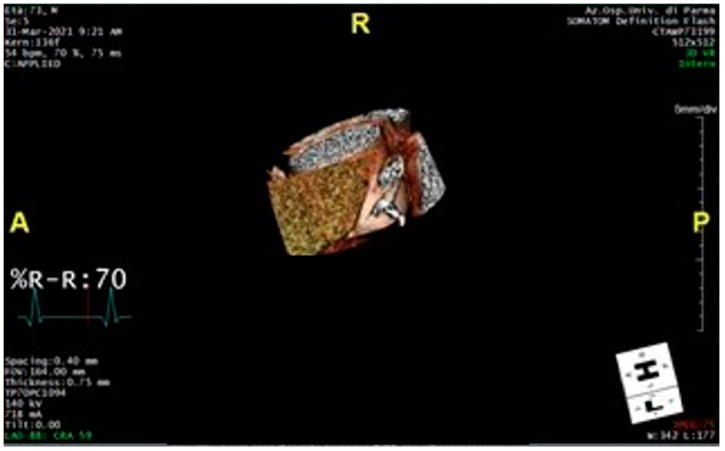
The figure shows the 3D reconstruction of the region of interest, i.e., the perivascular adipose tissue of the first 40 mm of the proximal ascending thoracic aorta (perivascular tissue with a density between −30 and −190 Hounsfield units).

**Table 1 diagnostics-15-00673-t001:** Comparison of the main characteristics between patients with Marfan syndrome (MFS) and the control group.

	All Patients (*n* = 97)	MFS Group (*n* = 54)	Control Group (*n* = 43)	*p* Values
**Demography and personal history**				
Age, years, median SD	43.4 (11)	41.5 (13)	45.9 (8)	0.058
Males, %	40 (41)	21 (39)	19 (44)	0.599
Hypertension, %	0 (0)	0 (0)	0 (0)	-
Diabetes, %	0 (0)	0 (0)	0 (0)	-
Obesity, %	0 (0)	0 (0)	0 (0)	-
Dyslipidemia, %	0 (0)	0 (0)	0 (0)	-
Smoking habit, %	0 (0)	0 (0)	0 (0)	-
CCS, %	0 (0)	0 (0)	0 (0)	-
**Chest computed Tomography**				
AA diameter, mm, median SD	33.2 (4.9)	33.4 (3.9)	33 (3.1)	0.695
PVAT in the AA [lower–upper quartile], HU	−72.6 [−74.8 to −70.9]	−70.6 [−72.6 to −68.5]	−75.1 [−77.1 to −73.1]	**0.002**

AA = ascending aorta; CCS = chronic coronary syndrome; HU = Hounsfield unit; MFS = Marfan syndrome; PVAT = perivascular adipose tissue attenuation; SD = standard deviation. *p* values < 0.05 are indicated in bold.

## Data Availability

The data presented in this study are available on request from the corresponding author due to ethical considerations.

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
