# Peer review of "Comparative Analysis of Perivascular Adipose Tissue Attenuation on Chest Computed Tomography Angiography in Patients with Marfan Syndrome: A Case–Control Study"

_diagnostics, 2025, doi:10.3390/diagnostics15060673_

Round 1
Reviewer 1 Report
Comments and Suggestions for Authors
The article summarizes the findings of a case-control study, which focused on the connection of peri-aortic adipose tissue attenuation and asymptomatic Marfan's syndrome in adult population. The introduction is well-written and I'd only recommend one extension: please give some insight on the importance of asymptomatic MFS screening. Is there any specific age when CTA and PVAT measurement should be performed?
Materials and Methods section is also well written. I'd like to highlight the importance of standardized imaging criteria, which is crucial for this type of attenuation measurement. In this study, this step was well designed, therefore I assume these findings should be reproducible.
I do have one question for the methods part: since the Aquarius WS software made automatic selections based on a pre-defined histogram, does this step carry a risk of selection error? Can the software confuse atypical PVAT with normal, other type of tissue? Consider include a scentence about this problem in the limitations part.
In results part, consider breaking down the results by gender to reveal any differences. If there are no differences, please consider providing this information in a sentence. Also, consider adding a figure, such as box and whiskers of the final results.
The discussion part is well-written, the referenced articles are recent and the section gives a thorough explanation of the relevance of findings in the field of MFS research. I'd recommend adding more insight on potential future prospective research, since the follow-up of these patients would provide evidence or counter-evidence to the hypothesis, that early changes in PVAT attenuation anticipate vascular disease.
Conclusions is well written, it summarizes the key ideas and message presented in this article in a compact style.
Author Response
Thank you very much for the review and for the highlighted suggestion. We have revised our manuscript and provided a point-by-point response in the attached file.

Reviewer 2 Report
Comments and Suggestions for Authors
The title and the abstract represent the content of the article very well. The article itself builds on their interesting case series (ref 30). The introduction describes the problem adequately and ends with a clear research question.
The methods describe the inclusion and exclusion criteria of both groups in a clear way. In inclusion period of 15 years, however, is quite long. Is the adaptation described in line 200 the result of an adaptation over time? The statistical analysis is adequate and the two groups are clearly comparable with respect to the demographic and clinical data.
The figures are clear and instructive.
The discussion is brief but adequate and the limitations are identified.
The clinical significance of the findings will be augmented by a follow-up (see line 316-317), not only with respect to these measurements, but especially with the clinical outcome (dilatation, dissection, need for surgery and survival). Is anything known about these patients with Marfan? A correlation of the degree of PVAT and future clinical events is of paramount importance. It would illustrate the predictive value of PVAT and identify patients needing a more thorough clinical follow-up.
Author Response
Thank you very much for the overall feedback and appreciation of our case-report study. Therefore, we have revised our manuscript and provided a point-by-point response in the attached file.
